# Honeydew Deposition by the Giant Willow Aphid (*Tuberolachnus salignus*) Affects Soil Biota and Soil Biochemical Properties

**DOI:** 10.3390/insects11080460

**Published:** 2020-07-22

**Authors:** Kyaw Min Tun, Andrea Clavijo McCormick, Trevor Jones, Stanislav Garbuz, Maria Minor

**Affiliations:** 1Wildlife and Ecology Group, School of Agriculture and Environment, Massey University, Private Bag 11222, Palmerston North 4410, New Zealand; k.m.tun@massey.ac.nz (K.M.T.); A.C.McCormick@massey.ac.nz (A.C.M.); s.garbuz@massey.ac.nz (S.G.); 2Plant & Food Research, Fitzherbert Science Centre, Batchelar Road, Palmerston North 4410, New Zealand; trevor.jones@plantandfood.co.nz

**Keywords:** *Tuberolachnus salignus*, honeydew, soil biochemical indicators, soil enzymes, yeasts, meso-fauna

## Abstract

Infestation of willow plants by the giant willow aphid *Tuberolachnus salignus* (Hemiptera: Aphididae) is associated with copious deposition of sugar-rich honeydew under the plant canopy. We explored the effect of aphid honeydew on the soil biota and biochemical indicators in a two-year field trial. Soil samples from under aphid-infested and control willow trees, as well as samples from black sooty mould spots under the aphid-infested willows were compared; soil samples before aphid inoculation were used as a baseline. The honeydew deposition had a positive effect on the total soil carbon (C), but not on the total soil nitrogen content or soil pH. Microbial biomass C, basal respiration, number of yeast colony forming units, and the geometric mean of activities for six enzymes were significantly higher in honeydew-affected soils than in the control treatment on both years. The honeydew deposition also increased soil meso-fauna abundance, especially in the black sooty mould spots. The soil biochemical properties, which differed before and after aphid infestation, showed considerable overlap between the first and second year post-infestation. The results highlight the cascading effects of *T. salignus* on soil biological activity and the importance of using a multitrophic approach to explore similar scenarios.

## 1. Introduction

The giant willow aphid, *Tuberolachnus salignus* (Gmelin) (Hemiptera: Aphididae), is an invasive stem-feeding pest of willow trees, which has recently arrived in New Zealand [1]. Willows (*Salix* spp.) are important multi-purpose farm trees used for biomass production, bioremediation, erosion control, and soil nutrient management [2,3]. As *T. salignus* is a new species in New Zealand, not much is known about the ecological consequences associated with its presence in willow growing systems, such as its effects on the soil biota.

One of the prominent features of infested willow plantings is the deposition of copious amounts of honeydew by aphid colonies, and the growth of black sooty mould on the leaves, stems, and on the soil surface ([1]; see also Figure 1). A single adult *T. salignus* can exude 1.71–2.08 mm^3^ of honeydew per hour [4,5]. Chemically, the honeydew is a mixture of water, carbohydrates (90–95% dry weight), amino acids (<5%), lipids and other nutrients [6,7]. When this energy-rich liquid is deposited on the leaves and understory plants, it is splashed onto the soil surface by rainfall [8]. It can be hypothesized that deposition of *T. salignus* honeydew on the soil surface will initiate a cascade of changes in soil processes, causing modifications in soil chemical properties, microbial activities and in the abundance of soil microbivores. Previous studies have linked the labile carbon (C) input from aboveground aphid herbivory to changes in belowground biochemical properties [9,10,11,12]. These effects are linked to aphid population density [13] and the identity of the aphid species [14].

Aphid honeydew deposition on the soil surface is expected to increase nutrient availability, fuel the growth of microbial communities in belowground systems [11,15], and influence the soil decomposition processes [16,17]. Microorganisms (bacteria, fungi, and other taxa) contribute to the functioning of soil ecosystems [18], regulating the processes of organic matter decomposition and nutrient cycling [19]. Soil microorganisms are assumed to be energy-limited, and as a result, mostly remain dormant when C resources are scarce [20]. The daily addition of sugar to the soil can cause a 2.5-fold increase in bacterial diversity, compared to control treatments, as sugar supplementation encourages the soil microbes to become active [21]. Aphid honeydew is a suitable growing medium for various saprophytic microbes [22] and has been shown to increase the activities of soil microorganisms [11,16]. Among soil microbes, yeasts are important degraders and saprotrophs [23,24], utilizing various C and nitrogen (N) sources [25]. Soil yeasts are ubiquitously present in many agroecosystems [26] and in nutrient-rich forest environments [23], and exhibit a quick response to changes in soil nutrient content [27].

Changes in soil microbial activity can also be reflected in the activities of soil enzymes [28]. Soil enzymes activity is a commonly used soil bioindicator [29], because of their quick response to subtle changes in available resources such as organic C input [30,31], and the ease of enzyme quantification [32]. So far, soil enzyme activities have not been used to explore honeydew-mediated changes in soil quality. Measuring the changes in the activities of soil enzymes following aphid infestation can provide a good tool to quantify the soil microbial responses to honeydew deposition [19,33].

An increase in microbial biomass could have potential consequences for soil meso-fauna, as their abundance is likely to be affected through food web interactions [14,34]. Soil meso-fauna (Collembola and Acari) live in top soil layers and play different functional roles in soil processes and nutrient cycling [35]. Collembola, Astigmata, and Oribatida are dominant soil microbivores [36,37,38,39,40,41], while Gamasida are mobile predators of meso-fauna [42]. Although some studies have been conducted to explore the effect of honeydew deposition on soil meso-fauna abundance [14,34], the results are inconclusive as both increases and decreases in meso-fauna abundance have been reported in honeydew-affected soils.

The aim of this study was to investigate the cascading effects of honeydew deposition by *T. salignus* on soil chemical properties (pH, C and N), soil microbial biomass and basal respiration, soil yeasts, abundance of soil meso-fauna (Acari and Collembola), and soil enzyme activity. Six enzymes were selected based on their sensitivities and importance in the electron transport system (dehydrogenase), cycling of C (glucosidase, invertase and amylase), and N (amidase and urease) in the soil. We analysed and compared soil biochemical properties and biota under control plants, under aphid-infested plants and in black sooty mould spots under aphid-infested plants. 

## 2. Materials and Methods

### 2.1. Willow Field Trial

A willow field trial with an area of 4000 m^2^ (50 m × 80 m) was established at the Orchard Block, Plant Growth Unit, Massey University, Palmerston North, New Zealand (40°22′41.70″ S, 175°36′30.67″ E, 30 m a.s.l). Average annual rainfall at the study site is 980 mm, ranging from 64 mm in the driest month (February) to 99 mm in the wettest month (July). Average annual temperature is 13.3 °C, fluctuating from 8.6 (July) to 18.1 °C (February) [43]. The soil type in the experimental area is a Manawatu fine sandy loam (Weathered Fluvial Recent Soil; [44]). Prior to planting willows, weeds were killed with glyphosate herbicide on 16 May 2017 and on 25 May 2017 the soil was rotary hoed in six rows. Each row was 1 m wide and 75 m long, with 4.0 m spacing between rows. The field trial was arranged in a split-plot layout, with three replicated blocks. Each block included two rows of willows; each row contained row plots of 12 ramets of fifteen willow cultivars. Willow cuttings (20 cm in length and 13 ± 2.6 mm in diameter) were planted on 16 June 2017, with 0.4 m spacing between cuttings within rows. After planting, the weeds were controlled by manual weeding and by spraying with Buster^®^ herbicide. The two treatments, presence of *T. salignus* and aphid-free control, were randomly allocated to the two rows within each block.

### 2.2. Aphids

Willow plants in the aphid-infested rows were inoculated with five adult aphids per plant on 25–27 January 2018 and 6–7 December 2019. Additional inoculations with ten adult aphids per plant were performed on 13–14 February 2018 and 30 January 2019. The willow plants in the control rows were inspected for colonising aphids on a weekly basis, and any aphids found were removed manually. The plants in control rows were sprayed with Mavrik^®^ insecticide on 28 February 2018 and on 17 January 2019, when manual control was impractical due to high population densities of *T. salignus*.

### 2.3. Soil Sampling

Soil samples were collected on the willow field trial site on 16 May 2017 prior to willow planting to assess spatial heterogeneity of the site. Following willow planting, samples were collected under the canopy of willow plants, in the 1.0 m wide cultivation zone, before aphid inoculation on 24 January 2018, after aphid inoculation on 22 June 2018, and on 2 July 2019. Three sampling points that were 20 m equidistant from each other, were marked along each of the six rows of the field trial (Appendix A).

Eighteen samples (one per sampling point), consisting of nine replicates from the aphid-infested and control rows, were collected during each sampling visit. Although all the plants in infested rows were inoculated with aphids, the honeydew was unevenly deposited, reflecting the distribution pattern of the aphid colonies. Therefore, additional soil samples were taken from black sooty mould spots (Figure 1) on the soil surface in the aphid-infested rows. Three samples (one per row) were collected on 22 June 2018, and six samples (two per row) were collected on 2 July 2019. Soil moisture content was measured three times at each sampling point using a TDR 300 Soil Moisture Probe (Spectrum Technologies Inc., Aurora, IL,****** USA), and the average of the three readings was then recorded. Soil temperature at 5 cm depth was measured with a QM7216 Digital Stem Thermometer. At each sampling point, two samples were collected, one for soil fauna extraction and another for analysing the soil chemical properties, microbial respiration and enzyme activities. The samples were put into labelled plastic bags, placed in an ice chest and immediately brought to the laboratory.

The samples used to quantify the soil meso-fauna (Collembola and Acari) were taken using a 25 cm^2^ soil corer to 5 cm depth. The sample (300–500 g) for soil chemical properties, microorganisms and enzymes was collected using a spade to 5 cm depth from five spots within a 1.0 m diameter circle around each sampling point, and then mixed thoroughly in a plastic tray to get a homogenous sample. Earthworms, plant roots, moss, stones, and other debris were removed before sieving the soil through a 2 mm drum sieve. The sieved soil was then divided into two subsamples. The subsample for analysing the soil chemical properties and enzyme activities was air-dried at room temperature, ground and sieved through a 1 mm mesh drum sieve. The subsample for determining the yeast colony forming unit (CFU), microbial respiration and biomass was frozen at −20 °C.

#### 2.3.1. Soil Chemical Properties

The soil pH was measured in a slurry containing 5 g of soil and 12.5 mL of distilled water, using an Orion Star™ A214 pH/ISE Benchtop Meter (Thermo Scientific, Waltham, MA, USA). The total C and N content were determined by Vario Macro Cube (Elementar Analysensysteme GmbH, Langenselbold, Germany) from the mixture of soil (75–100 mg) and tungsten oxide powder (25–50 mg).

#### 2.3.2. Soil Fauna

The soil samples for meso-fauna extraction were processed within 24 h. The fauna were extracted from the soil cores using a modified Berlese-Tullgren apparatus, as described by Oliver and Beattie [45]. Extraction was performed under 15 W light bulbs (Sylvania, OH, USA, 240–250 W) in a 17 to 30 °C temperature gradient in a temperature-controlled room for 7 days. The animals were collected into 70% ethanol and examined using an Olympus SZX12 stereomicroscope (Spach Optics Inc., Rochester, NY, USA). The Collembola were identified to an order level. The soil Acari were assigned to three suborders: Oribatida, Astigmata and Gamasida. The other meso- and macro-fauna, including small insects, spiders, centipedes, Isopoda, Diplura, Symphyla, annelid worms and Pauropoda, were grouped as “others”. The densities of the fauna were expressed as the number of individuals per m^2^.

#### 2.3.3. Soil Enzymes

The urease (EC 3.5.1.5), invertase (EC 3.2.1.26), β-amylase (EC 3.2.1.2), α-glucosidase (EC 3.2.1.20), dehydrogenase (EC 1.1.1.1) and amidase (EC 3.5.1.4) activities were assessed according to the protocols developed by Shcherbakova [46], Frankenberger and Johanson [47], Ross [48], Mfombep and Senwo [49], Serra-Wittling, et al. [50], Alef and Nannipieri [51] and Frankenberger [52], with slight modifications. Moist soil (1 g dry weight equivalent) was treated with 1.6 mL of triphenyltetrazolium chloride (TTC) before incubating at 30 °C for 24 h; 5 mL of acetone was added followed by incubation in the dark for 2 h to measure the dehydrogenase activity [51]. After incubating 0.25 g of soil with 2 mL of urea in phosphate buffer, and 20 µL of toluene at 37 °C for 4 h, the urease activity was assayed using a Genova Nano Micro-Spectrophotometer (Jenway, Stone, Staffordshire, UK) as the amount of nitrate released from urea [46]. The amidase activity was assessed using formamide substrate, and the amount of ammonia released during hydrolysis of the enzyme was measured at a wavelength of 400 nm in the above-mentioned spectrophotometer [52]. The invertase activity was estimated by measuring the amount of glucose and fructose released from sucrose, after incubating soil samples (0.3 g) with toluene and modified universal buffer at pH 5 [47]. The amylase activity was measured using starch solution as a substrate, according to Wainwright, et al. [53]. Determination of the α-glucosidase consisted of incubating soil samples (1 g) with toluene (0.2 mL), 67 mM sodium acetate buffer (4.3 mL, pH 5.0) and 50 mM maltose (0.5 mL) in plastic tubes at 37 °C for 1 h; the activity of this enzyme was assayed after placing the tubes in a boiling water bath for 5 min [49]. The activity of each enzyme was expressed based on 1 g of dry soil. One gram of each fresh soil sample was used to estimate the dry-weight equivalent conversion factor. The samples were oven-dried at 80 °C for 72 h until constant weight was achieved, and the dry weight measured.

The geometric mean of the enzyme activities (GMea) is regarded as a sensitive indicator for soil quality and soil health assessment [54,55,56]. The GMea calculation is based on the activities of all the assayed enzymes [54], and it is a more reliable index than any of the specific enzyme activities alone [57]. In this study, the GMea for six enzyme activities in the aphid-infested and control rows, and the black sooty mould soil spots was calculated according to Paz-Ferreiro, et al. [57] as follows:(1)$$\mathrm{the}\;\mathrm{GMea}=\sqrt[{6}]{\mathrm{Ure}\;\times \;\mathrm{Inv}\;\times \;\mathrm{Amy}\;\times \;\mathrm{Glu}\;\times \;\mathrm{Dehy}\;\times \;\mathrm{Ami}}$$
where Ure, Inv, Amy, Glu, Dehy and Ami represent urease, invertase, β-amylase, α-glucosidase, dehydrogenease and amidase, respectively.

#### 2.3.4. Yeasts

The frozen soil samples were incubated at 25 °C for 24 h. Fresh soil was divided into three plastic tubes (1 g dry soil equivalent each), and suspended in Milli Q water to obtain three dilutions (*v*/*w*, 1:5, 1:10 and 1:20) according to Yurkov, et al. [58]. After shaking the suspensions on an orbital shaker for 1 h, 0.1 mL aliquots were plated in triplicate on casein-peptone glucose yeast extract agar, supplemented with chloramphenicol (0.1 g L^−1^). Lactic acid was added to acidify the medium to pH 4.5. The plates were incubated at 25 °C for 2 days and then transferred to a chiller (5 °C) to prevent mould development. Visible colonies were counted weekly for three consecutive weeks. The yeast counts were expressed as the colony forming units (CFU) per 1 g of dry soil, multiplied by the dilution factor (5, 10 and 20).

#### 2.3.5. Microbial Properties

The microbial biomass C (Cmic) and basal respiration (BR) were determined by the substrate-induced respiration (SIR) method [59]. The frozen soil samples were incubated for 24 h at 25 °C. The samples (1 g dry weight equivalent) were weighed and placed into 22 mL glass vials. After dropwise addition of 0.1 mL glucose solution (8 mg g^−1^ soil), the vials were closed with airtight lids containing a septum in the centre. The vials were incubated at 22 °C for 3–5 h. Air samples were collected using a syringe, and then injected into the CO_2_ analyser (HP 3396 Series II Integrator, Hewlett Packard, Palo Alto, CA, USA). The Cmic (μg C g^−1^ soil) was calculated as: Cmic = SIR (μL CO_2_ g^−1^ soil h^−1^) × 40.04 + 0.37, according to Anderson and Domsch [60]. A similar procedure was used for determining the BR, but only 0.1 mL of water was added to the vials prior to the 24 h incubation period at 22 °C. The BR of the soil samples was measured as μg CO_2_-C g^−1^ dry soil h^−1^. The microbial metabolic quotient (qCO_2_) was calculated by dividing the BR by the Cmic, and expressed in μg CO_2_-C mg^−1^ Cmic h^−1^ [61].

### 2.4. Data Analysis

All analyses were performed in R version 3.5.1 [62]. A Shapiro–Wilk test was used to check whether the data distributions met the assumption of normality. Generalized linear models (GLMs) were used for analysis of the treatment effects on the soil chemical and microbial properties, and enzyme activities. The gamma distribution with log-link function was used for non-normally distributed variables, while the normally distributed data were fitted using the gaussian distribution with identity link. Count data (meso-fauna and yeast CFU counts) were analysed in the GLM using the Poisson and quasi-poisson error distributions. The GLMs were fitted separately for the two sampling times, after the willow plants were inoculated with aphids. The pre-treatment sampling on 24 January 2018, prior to the aphid inoculation, was used as the baseline measurement, but was not included in the analysis of the aphid infestation treatment vs. control. The library “*multcomp*” was used for multiple comparisons using Tukey’s HSD test, whenever the GLM results showed global significant differences of the means. Full results of all GLM tests are provided in the Appendix A.

The principal component analysis-linear discriminant analysis (PCA-LDA) was used to visualise the data and maximize the treatment segregation [63,64]. All variables were square-root transformed, scaled, and centred (divided by their respective standard errors) to assure equal variance, before conducting the PCA-LDA [65]. First, the PCA was performed to produce the principal components (PCs) in the “*FactoMineR*” and “*factoextra*” packages. The first eight PCs together explained more than 80% of the variance in the data and were used in the linear discriminant analysis (LDA). The “*caret*”, “*MASS*” and “*tidyverse*” packages [63] were used to perform LDA.

In the PCA-LDA evaluating the honeydew-related changes in the soil biochemical properties among the control, aphid-infested treatments, and black sooty mould spots, the baseline measurements (collected before aphid inoculation) were excluded, as the aphid absence on the plants meant no honeydew was deposited on the soil surface. In the PCA-LDA, which compared the selected soil indicators over time, the baseline measurements as well as data from first and second year after aphid inoculation were included. Soil temperature and moisture measurements were excluded from the time analysis to remove bias due to weather.

In the current study, the effect of the willow cultivars was excluded from consideration. Previous research had showed the total sugar content in the honeydew of *T. salignus* was not statistically different among the willow cultivars [66]. The soil samples were taken from under the canopies of the same willow cultivars, in the aphid-infested and control rows.

### 2.5. Structural Equation Modelling

Structural equation modelling (SEM) is a more reliable approach than univariate correlations and regressions, because it provides path coefficients to examine the multiple associations in a multi-layered system [67]. SEM was constructed to explore how honeydew deposition could influence soil biochemical processes and functions, and to quantify the relative contribution of the different variables, which form a network of causal relationships [68].

The *a priori* hypothetical model was first constructed to describe the causal relationships for the effects of aphid honeydew on the soil environment, linking *T. salignus* honeydew deposition to Cmic, specific enzyme activities, meso-fauna abundance, and the geometric mean of the enzyme activities (GMea). This was based on a modification of the path model constructed by Milcu, et al. [14]. Variables for the model were selected based on the PC scores (Appendix A) and previous literature. We then used SEM to calculate the coefficients associated with each path in SPSS Amos 25 (IBM, Armonk, NY, USA) [69].

The data on the soil analysis from the two sampling dates after aphid inoculation were pooled for this analysis; the pre-treatment data, prior to aphid inoculation, were excluded. The honeydew input was coded as a categorical (ordinal) variable, with 0 for control, 1 for aphid-infested, and 2 for black sooty mould spots. The Cmic, total C and N contents were selected to evaluate the direct effect of honeydew deposition. The Cmic was chosen to estimate the indirect effect of honeydew deposition on the Gmea and meso-fauna abundance, as Cmic has been shown to associate with soil fauna [14], soil chemical properties (total C, total N) and enzyme activities [70].

The critical ratio of the multivariate kurtosis and squared Mahalanobis distance were checked for multivariate normality and the presence of outliers [71]. The pooled dataset, containing the selected variables over the two sampling times, was square-root transformed to meet the assumption of multivariate normality. The maximum likelihood estimation was used to test the path coefficients in the SEM models. Standardized coefficients were calculated for all the variables in the paths diagram [72,73]. The Chi-square test, probability value of the likelihood ratio test, comparative fit index (CFI), root mean square error of approximation (RMSEA), and Akaike’s information criterion (AIC) were used to evaluate the model fit.

## 3. Results

### 3.1. Soil Chemical Properties

Before the willows were planted, the soil of the field trial site had a mean pH value of 6.1 ± 0.05, 2.4 ± 0.08% total C and 0.25 ± 0.01% total N. None of these parameters exhibited any significant spatial differences prior to willow planting. The baseline measurements for soil chemical properties after planting prior to aphid infestation are included in Table 1.

Honeydew deposition resulted in a higher total C content in the black sooty mould spots compared to the control and aphid infestation treatments in both first and second year after aphid infestation (Table 1). There was no effect of the treatments on the soil pH values or total N content in both years. The C:N ratio was significantly higher in the black sooty mould spots in both years (Table 1).

### 3.2. Soil Meso-Fauna

Of the 4.80 × 10^6^ soil meso-fauna collected in samples, Collembola was the most dominant taxon, comprising 75.5% of the total. Significantly higher Collembola densities were observed in the black sooty mould spots in the first year, and in both the aphid-infested rows and black sooty mould spots in the second year, compared to the control treatments (Figure 2a).

The soil Acari (0.92 × 10^6^ individuals) accounted for 19.3% of the soil meso-fauna. Gamasida was the most abundant taxon, comprising 67.9% of the Acari. The aphid infestation had a significant effect on gamasid mites only in the first year, with higher densities in the black sooty mould spots (Figure 2b). Astigmata accounted for 19.3% of the total soil mites. Their densities were higher in the black sooty mould spots than the control and aphid infestation treatments in first year (Figure 2c). Oribatida was the least abundant group (12.8%) of soil mites. In the second year, significantly higher densities of Oribatida were recorded in the black sooty mould spots than in aphid infestation treatment but not in the control treatment (Figure 2d). The black sooty mould spots also had higher population densities of the ‘other’ fauna in the second year (Figure 2e).

### 3.3. Soil Enzymes

In general, honeydew deposition affected the soil enzyme activities, which tended to be higher in black sooty mould spots than in aphid-infested and control treatments (Figure 3). The dehydrogenase and β-amylase had significantly higher activities in the black sooty mould spots in both years; in the second year the activity of these enzymes was also higher in the aphid infestation treatment than in the control (Figure 3a,d). The soil urease activity was consistent across the two years and showed a significant response to honeydew deposition in the order: black sooty mould spots > aphid-infested > control (Figure 3b). Both the amidase and invertase had significantly higher activities in the black sooty mould spots in the first year; in the second year the trend remained, but the differences were not significant (Figure 3c,e). The activity of soil α-glucosidase was significantly higher in black sooty mould spots than in the control treatment in both years, but there was no difference between the control and aphid-infested treatments (Figure 3f). Similar to the specific enzyme activities, the Gmea for the six enzymes was significantly influenced by the honeydew deposition; the Gmea in the first and second year was in the order: black sooty mould spots > aphid-infested > control (Figure 4).

### 3.4. Soil Microbial Properties and Yeast CFU

In both years, soil Cmic (Figure 5a), basal respiration (Figure 5b) and yeast CFU (Figure 6) increased in the order: control > aphid-infested > sooty mould spots, with all differences being significant. The microbial quotient (qCO_2_) was significantly higher in the black sooty mould spots than in the aphid-infested and control treatments only in the second year (Figure 5c). 

### 3.5. Principal Component Analysis-Linear Discriminant Analysis (PCA-LDA)

The PCA-LDA showed the localized effect of the honeydew deposition in the black sooty mould spots, and the change after aphid infestation (Figure 7). 

The first PCA-LDA clearly separated the black sooty mould spots from the other two treatments, but there was some overlap between the control and the aphid-infested treatment (Figure 7a). For the three treatments, the first (LD1) and second (LD2) discriminant functions explained 95.8 and 4.2% of the total variability, respectively. The GMea, BR, Cmic, C:N ratio, urease, β-amylase, qCO_2_, and yeast CFU were the variables that contributed the most to the separations along the PC1 (Appendix A), which had the highest discriminant coefficient in the first linear discriminant (LD1).

The second PCA-LDA clearly separated the sampling times prior to aphid inoculation (before aphid infestation), and after aphid inoculation in the first and second years, but there was considerable overlap between the first and second year post-infestation (Figure 7b). The GMea, BR, Cmic, C:N ratio, urease, β-amylase, qCO_2_, and yeast CFU were also the variables that contributed the most to the separation along PC1 (Appendix A) which had highest weight in separating sampling times in LD1 (84.9% of the total variability) (Figure 7b).

### 3.6. Structural Equation Modelling (SEM)

The SEM for the soil enzymatic response revealed a fairly good fit to the data (χ^2^ = 125.7, df _(66–28)_ = 38, *p* < 0.001, RMSEA = 0.00, CFI = 1, AIC = 181.7). The honeydew deposition had a positive direct effect on the total soil C and microbial biomass C (Cmic), but not on the total N (Figure 8a). Total soil C also increased the Cmic, while the total N had significant negative effect on Cmic. The Cmic increased the activities of all six assayed enzymes (Figure 8a), but the degree of influence was largest for β-amylase, urease, and dehydrogenase. The increased activities of urease, α-glucosidase and invertase contributed most to the geometric mean of the enzyme activities (GMea).

The SEM for the soil meso-fauna abundance had a better fit to the data (χ^2^ = 87.3, df_(36–19)_ = 17, *p* < 0.001, RMSEA = 0.00, CFI = 1, AIC = 72.0). The honeydew deposition increased the Cmic, which increased the abundance of the Collembola and Astigmata, but no significant effect was found for Oribatida mites (Figure 8b). The abundance of Gamasida was positively correlated with the abundance of their prey—Collembola, Astigmata and Oribatida.

## 4. Discussion

In the willow field trial, the development of black sooty mould spots on the soil surface (Figure 1) is an indication of a high population density of *T. salignus*. As expected, we found that the copious deposition of honeydew on the soil surface affected the soil biological and biochemical properties, especially in the spots marked by the black sooty mould. Overall, the soil biological indicators (microbial properties, enzyme activities and meso-fauna abundance) were found to be more sensitive to aphid honeydew deposition than the soil chemical properties.

In our study, *T**. salignus* honeydew deposition increased the soil total C content but did not change soil total N content. Stadler, et al. [16] found that honeydew input increased the dissolved organic C in litter, but reduced inorganic N content, and suggested that net N immobilization had occurred. Aphid honeydew is a C-rich but N-poor resource [5], inducing the population of soil microorganisms to increase and then compete for the limited N, which can increase the N immobilization rate and result in the depletion of inorganic N [16,74]. Thus, the honeydew deposition could indirectly decrease the soil N content through enhanced microbial activity [15,75], where microorganisms could emerge as potential competitors of willow plants for nitrogen resources [76]. However, there is some evidence that N limitations can be compensated through increased non-symbiotic N_2_-fixation by soil microorganisms [77].

The results of our study showed that sugar supplementation in the honeydew increased the yeast CFU count, microbial biomass C and microbial respiration. Soil microorganisms are mostly energy limited and remain dormant in the absence of a suitable substrate [20], so honeydew addition could increase their population numbers and respiration rate by promoting favourable growth conditions [78]. Soil yeasts prefer nutrient-rich environments and are known to utilize low molecular weight sugars [23] that are the major components of aphid honeydew. The increase in microbial biomass C (Cmic) and basal respiration are in accordance with the study of Milcu, et al. [14], who found that Cmic and basal respiration increased by 330% and 58.4%, respectively, in honeydew treatments. However, care should be taken in interpreting the microbial response to honeydew addition, as the sugar component of honeydew can shut down the metabolism of some microbes [79]. Further studies using molecular techniques are advised to determine the changes in the soil microbial community structure following honeydew deposition.

The activity of soil enzymes is regarded as a direct measurement of the metabolic response of the soil microbial communities to nutrient availability [28]. Our results show a significant effect of nutrient supplementation from aphid honeydew on the soil enzyme activities. Although honeydew contains an unbalanced ratio of C to N, we found that both C-hydrolysing (β-amylase, invertase, α-glucosidase), and N-hydrolysing (urease and amidase) enzymes positively responded to the honeydew deposition. Dehydrogenase was also found to be a sensitive indicator of increased microbial activity [80,81] as a result of the supplementary C input from honeydew. The interpretation of enzyme activity results should be treated cautiously as enzyme assays generate the highest potential estimates, under optimum substrate, pH and temperature conditions, rather than the actual values [51]. However, both the specific enzyme activities and the GMea were suitable indicators discriminating the black sooty mould spots from the control, with GMea being the best predictor. 

Soil meso-fauna (Collembola and Acari) normally live in the topsoil layers, and play different functional roles in the soil processes and nutrient cycling [35]. We observed that Collembola counts in the honeydew-affected soil were higher than in control. Sinka, et al. [34] found no significant influence of honeydew deposition on Collembola abundance, while Milcu, et al. [14] reported the decline in Collembola and mite numbers in soil treated with synthetic honeydew. Seeger and Filser [82] noted that the effect varied for different collembolan taxa. The mite groups Astigmata and Oribatida are fungivores and saprophages [83], while the Gamasida are predators of other soil mites and Collembola [84]. In our study, the abundance of Oribatida and Gamasida varied over time, while that of Astigmata was fairly consistent. All three mite groups are known to respond to external resources [85,86], with the degrees of response to the aphid honeydew reflecting their different life histories. The Astigmata are known for their rapid response to the changing environment, due to their faster metabolism, shorter generation time, and higher fecundity than the Oribatida [83]. On the other hand, the higher population density of Gamasida reflects the presence of the prey on which they feed [87].

The SEMs were a useful tool to assess the multiple linkages between the honeydew and the soil biological and biochemical indicators, linking the aboveground herbivory to the below-ground soil processes. The path diagrams show that honeydew deposition by *T. salignus* has a multitrophic cascading effect on the soil biota, similarly as in Michalzik, et al. [13], Milcu, et al. [14] and Stadler, et al. [88]. In the current study, the Cmic was positively correlated with the soil total C content, but not with the total N. Our results are in line with those of Cheng, et al. [70] and Johnson, et al. [89], who suggested that the Cmic was mainly dependent on the soil C source, and additional N input could decrease the Cmic.

In our study system, the willow plants, aphids, soil chemical properties and soil biota are interacting with each other, highlighting the need for a multitrophic approach to investigate similar settings. It is important to mention that this study was conducted over two years only, so the long-term effects of the honeydew deposition on the soil remain unknown and require further investigation. Likewise, the impacts of the other sources of variation, including the different insect and host plant species [14], weather conditions [90] and seasonality [91], require additional attention.

## 5. Conclusions

The deposition of *T. salignus* honeydew affect the various soil biotic and abiotic properties through a multitrophic cascade. The aphid honeydew provides an energy-rich source for the soil microbes, causing an increase in the Cmic, that leads to increased soil enzyme activities. These processes affect the abundance of soil meso-fauna microbivores and their predators. This example illustrates the importance of multitrophic interactions, and the cascading effects of an aphid herbivore on soil chemical properties and soil biological communities.

## Figures and Tables

**Figure 1 insects-11-00460-f001:**
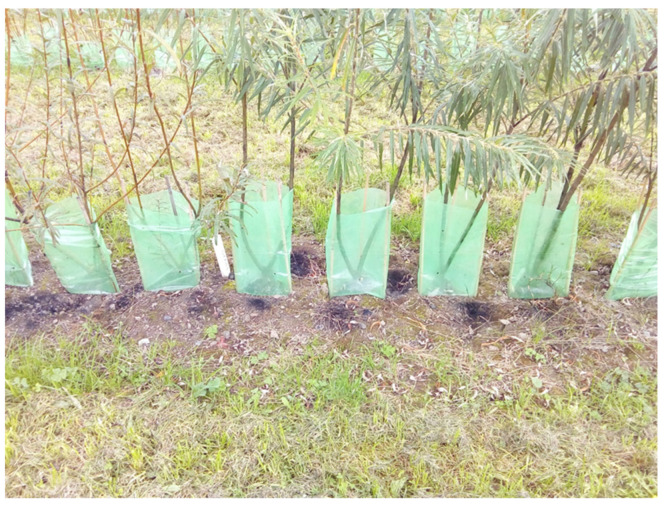
Black sooty mould spots under the canopy of willow plants.

**Figure 2 insects-11-00460-f002:**
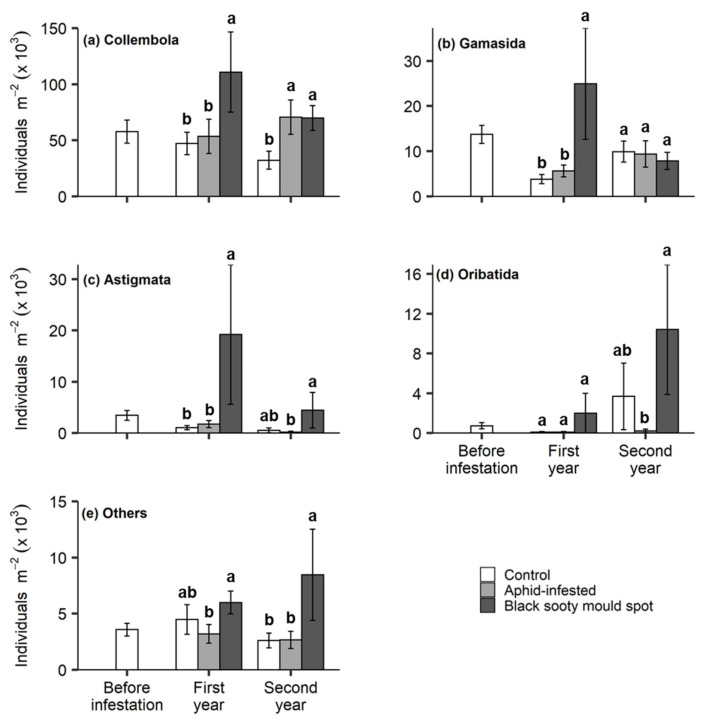
The effect of *T. salignus* honeydew deposition on the abundance of soil meso-fauna: Collembola (**a**), Gamasida (**b**), Astigmata (**c**), Oribatida (**d**) and others (**e**), prior to aphid inoculation (before infestation), and after aphid inoculation (control, aphid-infested, and black sooty mould spots) in the first and second year. The values are the means ± SE. Different letters indicate statistically significant differences between treatments within each sampling time (Tukey’s HSD test, α = 0.05).

**Figure 3 insects-11-00460-f003:**
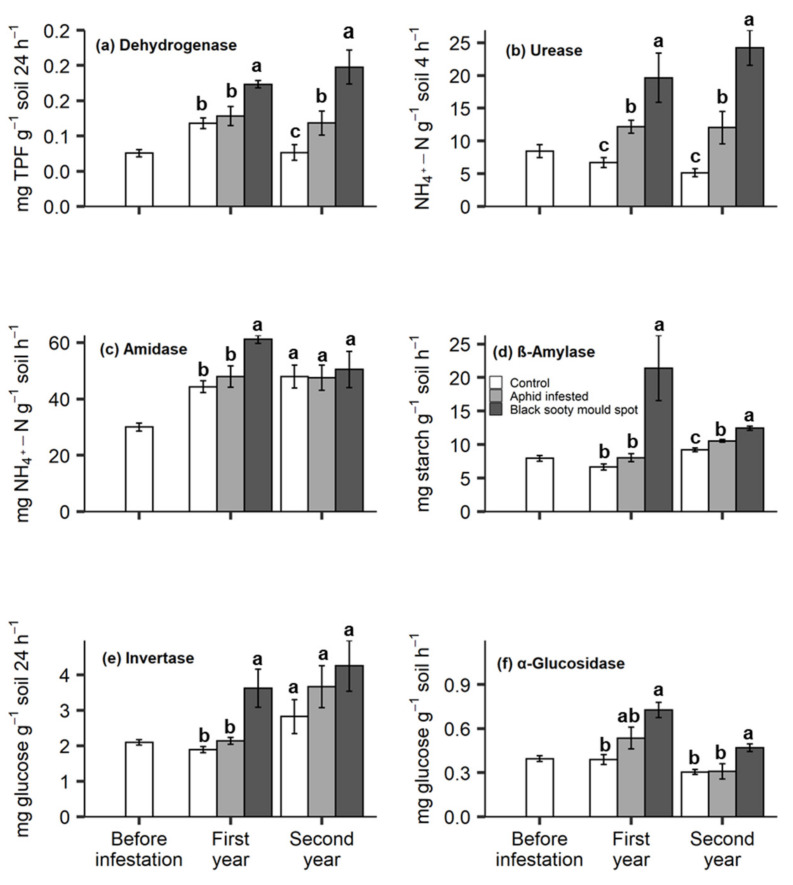
The activity of the soil enzymes: dehydrogenase (**a**), urease (**b**), amidase (**c**), β-amylase (**d**), invertase (**e**) and α-glucosidase (**f**) under the canopies of willow plants, prior to aphid inoculation (before infestation), and after aphid inoculation (control, aphid-infested, and black sooty mould spots) in the first and second year. The values are the means ± SE. Different letters indicate significant differences between the treatments at each sampling time (Tukey’s HSD test, α = 0.05).

**Figure 4 insects-11-00460-f004:**
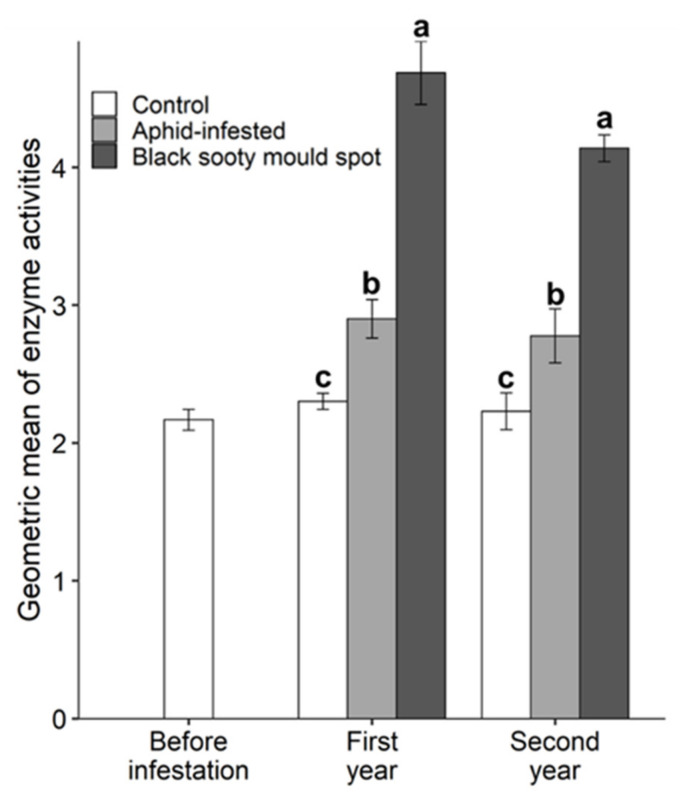
The effect of *T. salignus* honeydew deposition on the geometric mean of soil enzyme activities, prior to aphid inoculation (before infestation), and after aphid inoculation (control, aphid-infested, and black sooty mould spots) in the first and second year. The values are the mean ± SE. Different letters represent statistically significant differences between the treatments at each sampling time (Tukey’s HSD test, α = 0.05).

**Figure 5 insects-11-00460-f005:**
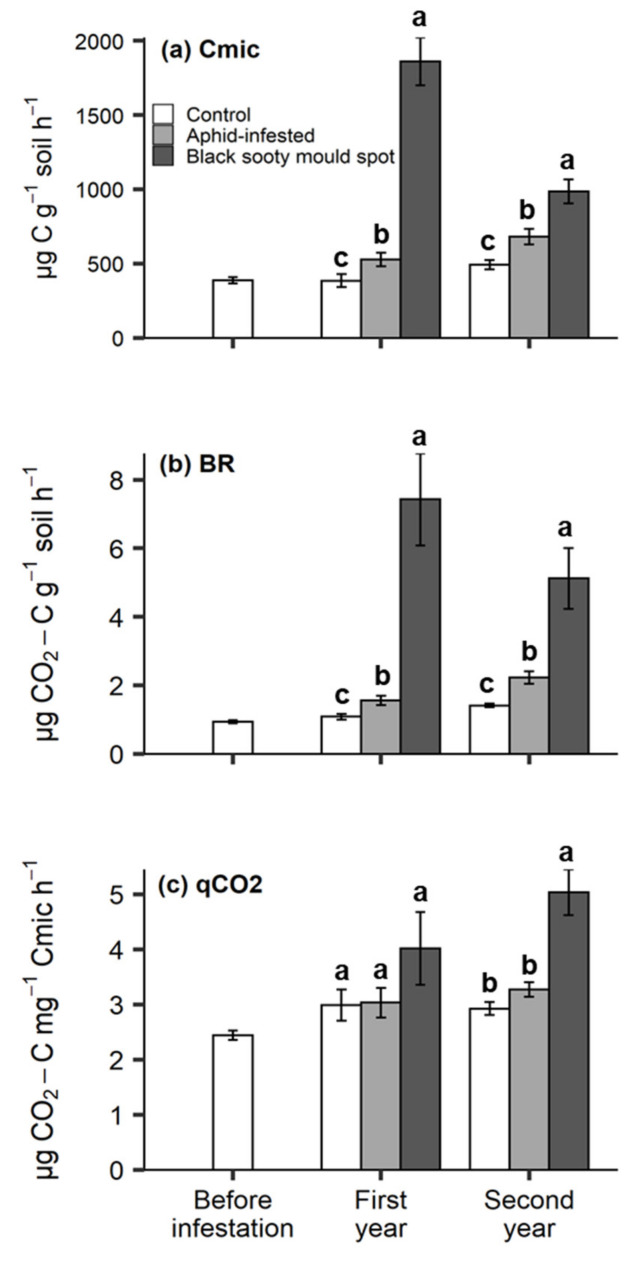
The effect of *T. salignus* honeydew deposition on (**a**) soil microbial biomass (μg C g^−1^ soil), (**b**) basal respiration (μg CO_2_-C g^−1^ soil h^−1^), and (**c**) metabolic quotient over time (μg CO_2_-C μg^−1^ Cmic h^−1^), prior to aphid inoculation (before infestation), and after aphid inoculation (control, aphid-infested, and black sooty mould spots) in the first and second year. The values are the means ± SE. Different letters indicate statistically significant differences at each sampling time (Tukey’s HSD test, α = 0.05).

**Figure 6 insects-11-00460-f006:**
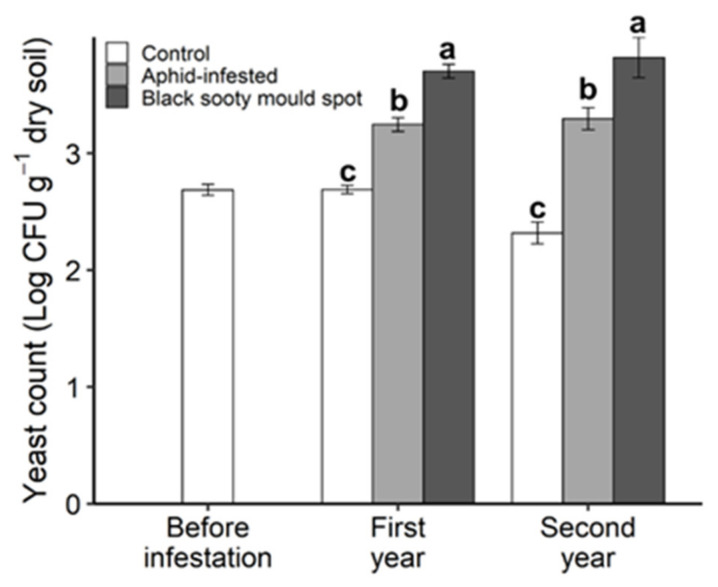
The yeast colony forming units (CFU) per gram of dry soil, collected under the canopies of willow plants prior to aphid inoculation (before infestation), and after aphid inoculation (control, aphid-infested, and black sooty mould spots) in the first and second year. The values are the means ± SE. Different letters indicate statistically significant differences at each sampling time (Tukey’s HSD test, α = 0.05).

**Figure 7 insects-11-00460-f007:**
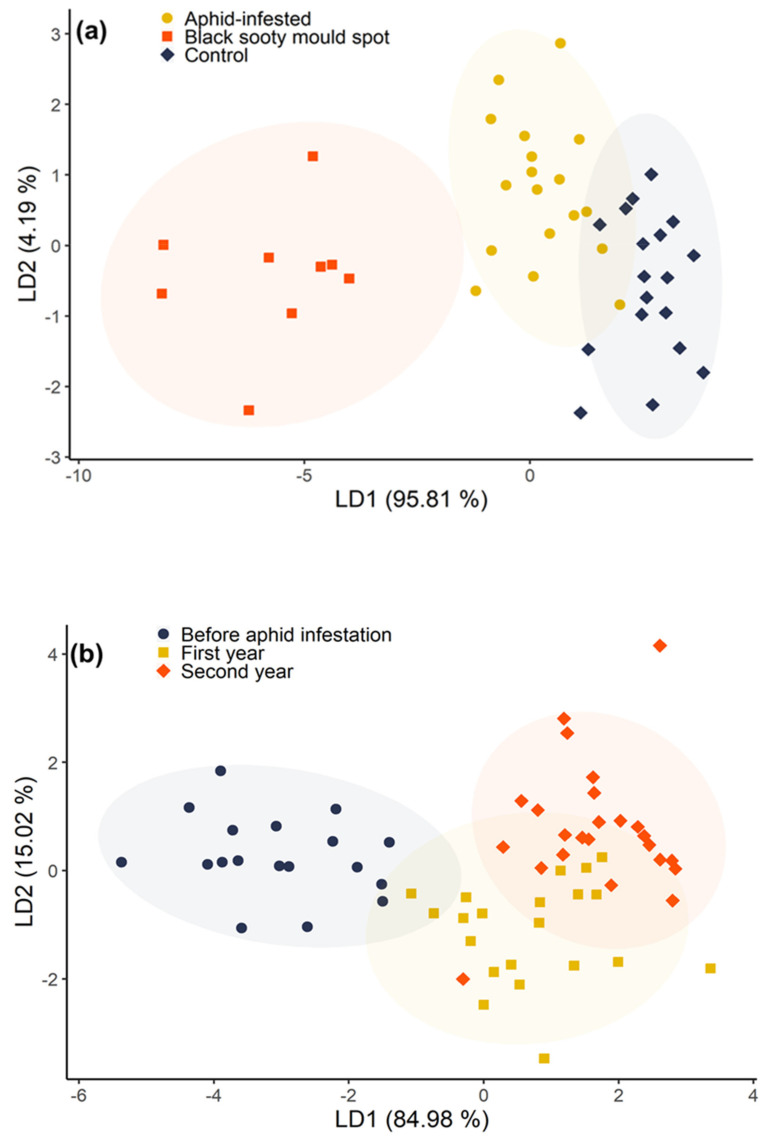
PCA-LDA bi-plots of the soil biochemical variables, classified (**a**) by treatment, and (**b**) by sampling time. A PCA was run to reduce the dimensions, followed by LDA to separate the treatments and sampling times. The variables were scaled and centred prior to the analysis. The shaded ellipses represent the 95% confidence areas.

**Figure 8 insects-11-00460-f008:**
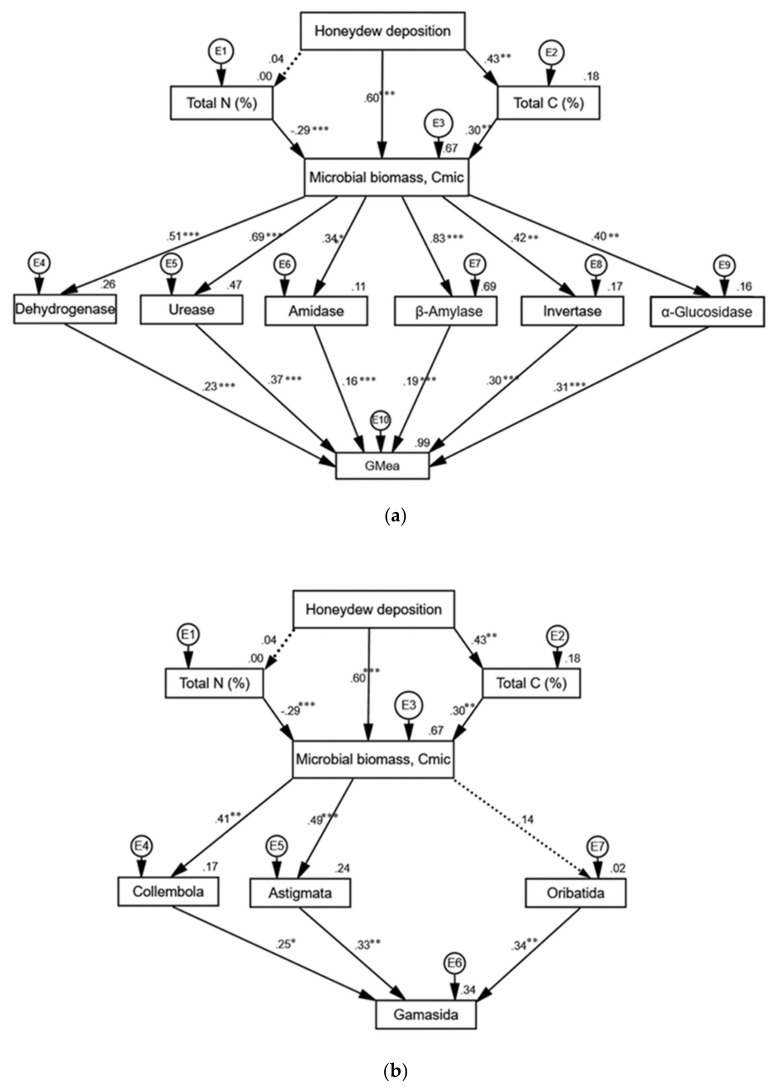
Path diagrams for the effects of *T. salignus* honeydew deposition on (**a**) soil biochemical properties, and (**b**) meso-fauna communities. The circles above the rectangles indicate the error terms. The solid and dotted arrows represent significant and non-significant associations, respectively. The path coefficients are the standard regression weights, with asterisks showing different levels of significance (* *p* < 0.05, ** *p* < 0.01, *** *p* < 0.001). The squared multiple correlations (r^2^) are expressed above the top right corner of each rectangle.

**Table 1 insects-11-00460-t001:** Soil chemical properties before aphid inoculation (baseline) and after aphid inoculation (control, aphid-infested and black sooty mould spots), during the first and second year of the experiment. Values are the means ± SE. Different letters in each column indicate significant differences between the treatments at each sampling time (Tukey’s HSD test, α = 0.05).

		Sampling Time		
Parameter	Treatment	Before Aphid Infestation	First Year	Second Year
pH	Baseline	5.75 ± 0.03		
	Control		6.102 ± 0.032a	6.073 ± 0.029a
	Aphid-infested		6.177 ± 0.052a	6.058 ± 0.065a
	Black sooty mould spots		6.163 ± 0.091a	6.067 ± 0.040a
Total C (%)	Baseline	2.40 ± 0.06		
	Control		2.146 ± 0.076b	2.278 ± 0.110b
	Aphid-infested		2.260 ± 0.061b	2.297 ± 0.084b
	Black sooty mould spots		2.490 ± 0.023a	2.547 ± 0.070a
Total N (%)	Baseline	0.27 ± 0.01		
	Control		0.238 ± 0.006a	0.247 ± 0.011a
	Aphid-infested		0.249 ±0.005a	0.242 ±0.008a
	Black sooty mould spots		0.237 ± 0.009a	0.247 ± 0.006a
C:N	Baseline	8.79 ± 0.13		
	Control		9.008 ± 0.136b	9.207 ± 0.178b
	Aphid-infested		9.076 ± 0.107b	9.483 ± 0.216b
	Black sooty mould spots		10.547 ± 0.306a	10.302 ± 0.140a

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
