# Peer review of "Honeydew Deposition by the Giant Willow Aphid (Tuberolachnus salignus) Affects Soil Biota and Soil Biochemical Properties"

_insects, 2020, doi:10.3390/insects11080460_

Round 1

Reviewer 1 Report

The revised version of the manuscript is largely improved (mainly Materials and methods section). However, some issues still remain to be solved and others have emerged.

Throughout the manuscript multiple terms are still used for one thing (e.g., bioprocesses x soil biochemical properties x soil chemical properties x soil properties, microflora x microorganisms and others). Select one term and then use it consistently. Without that, the text is not comprehensible enough. There is no rationale for using mixed Latin and common names for specific groups (i.e. mites x Acari). When you use Collembola (i.e. Latin name), use Acari for mites.

Abstract

Line 22: Delete “(N)“ as it is no further used.

Introduction

Although new references have been included, no knowledge gaps have been specified. Although you state that some changes are hypothesized, you refer to studies that investigated them in the next paragraphs (line 51: is is not expected, it was found based on references). What are the knowledge gaps you wanted to fill in? What are your specific hypotheses that were tested?? How does your manuscript differ from those that have already been published? Maybe focus on a different ecosystem studied, i.e. willow plants on New Zealand. By the way, no mention on this specific ecosystem in the introduction has been made. Why have you selected this ecosystem and what is so special to study it?

Line 40: Jílková et al. 2020a only focus on soil chemical properties, while Jílková et al. 2020b focus on soil biological properties. Change the reference.

Jílková et al. (2020b) Organic matter decomposition and carbon content in soil fractions as affected by a gradient of labile carbon input to a temperate forest soil. Biol Fertil Soils 56: 411-421.

Materials and methods

Line 97: delete “that“

Lines 120-123: When you do not provide data on soil moisture and temperature, delete the methods as well.

Line 134: There is no need to store air-dried samples at 4 °C. They are dry already and no further changes can occur there.

Line 135: “... microbial respiration and biomass“

Line 211: “... microbial properties measurement“ (or a similar term consistent with the rest of the manuscript), but not microbial biomass.

Results

Lines 269-271: Here you refer to biochemical and biological parameters but where are the results shown? Did you measure them at all? Change the sentence to: “The baseline measurements for soil properties after planting prior to aphid infestation are included in Table 1.“

Lines 274-278: Higher than what? You always need to compare with something.

Statistical results are not shown anywhere! Only results of the post-hoc tests. Provide complete results for all statistical tests done (i.e. F, df, P...).

Lines 288-292: “... higher densities of Oribatida were recorded in the black sooty mould spots than in aphid infestation treatment but not in the control treatment.“ The same situation applies to Astigmata.

Line 310: The results showed, not suggested.

Lines 311-314: This sentence is not comprehensible. Try to restructure it.

Tables and figures

Table 1: Title: “Soil chemical properties...“

Figure 2 and 3: Delete “All graphs have the same x-axis“ and “the untransformed“.

Figure 3 and 5: Easier and more comprehensible would be to include the units to the y-axis.

Discussion

Go one more time critically through the discussion section (or maybe let it read to somebody) as some statements are vague, do not make sense or are not true at all.

Line 353-362: Here you discuss total N, what about total C? Jílková et al. 2012 did not investigate total C and N, only pH. Moreover, Jílková et al. 2020a only found differences in total C and N in the subsoil mineral horizon, not surface what you sampled. This paragraph does not make much sense. Why could the difference between your results and results of Stadler et al. be caused by C:N ratio of the honeydew when probable very similar honeydew with very similar C:N ratio was used in both studies? Moreover, you did not measure specific C:N ratio of your honeydew to interpret the differences between your and Stadler et al. results. What host plants?

Line 364: “... microbial respiration and biomass“.

Line 376: Change the sentence to: “... as a direct measurement of the metabolic response of the soil microbial communities to nutrient availability.“

Line 381: No relationship was determined in your study or provide a reference.

Lines 382-383 and 423: Microbial activity and biomass are not closely connected as microbial activity can be increased without increasing microbial biomass (Blagodatskaya and Kuzyakov 2013).

Blagodatskaya, E., Kuzyakov, Y., 2013. Active microorganisms in soil: Critical review of estimation criteria and approaches. Soil Biology and Biochemistry 67, 192-211.

Line 392: Why should there be any negative effect? This information comes from the blue. Try to restructure the sentence or the whole paragraph.

Line 407: ”… The path diagrams show that honeydew deposition by T. salignus has a multitrophic cascading effect on the soil biota, similarly as in [6,7,90].“

Lines 409-412: If the N content is very low in honeydew, then I would rather expect no effect, not negative. This does not make sense.

Line 413: “... soil chemical properties and soil biota are interacting...“

Line 416: Here, the soil health is mentioned for the first time. How is it defined?

Reviewer 2 Report

The authors followed my suggestions in rewiewing the manuscript; there is only a small correction about references to do, as notated in the attached pdf (page 16, lane 411). Then the manuscript will be acceptable for publication.

Author Response

Reviewer's comments: References should be numbered as in the rest of the text. Please, control all the text regarding the number of references.

Authors' reply: All references were numbered throughout manuscript. Thank you.

Round 2

Reviewer 1 Report

The manuscript has largely improved and I agree with its publication.

This manuscript is a resubmission of an earlier submission. The following is a list of the peer review reports and author responses from that submission.

Round 1

Reviewer 1 Report

The manuscript “Honeydew deposition by the giant willow aphid (Tuberolachnus salignus) affects soil biota and biochemical properties“ describes clearly a positive effect of honeydew deposition on the soil properties. Although it is clearly written and the data are presented correctly, I have some comments which need to be addressed and resolved before the manuscript can be accepted for publication.

General comments

I miss comparisons of the results before planting with after planting and after infestation. Although you state that you compared those data in the abstract (line 19), you did not. You only presented the data without any statistical comparison or comments on the results. Moreover, comparisons of the 2 consecutive years are missing. Repeated measures ANOVA should have been used to test that.

Overall the literature used is pretty old although new publications on the honeydew deposition and its effect on soil chemical and biological properties have emerged recently (Jílková et al. 2012, 2018, 2020a,b; Domisch et al. 2009 etc.). Refer to them and then use them to interpret your results.

I have severe problems with your using of herbicides and insecticides. It must have affected the soil under the willow seedlings and changed your results. Have you tested that? Moreover, insecticides were not used on aphid-infected trees which might have caused differences between aphid-infested and control treatments.

I have problems with processing of soil samples for measurements (i.e. air drying or freezing). The properties measured after those treatments must have been affected (as referred to below).

Specific comments

Throughout the manuscript multiple terms are used for one thing (e.g. mesofauna x microarthropods x soil fauna, soil mites x Acari x mites, microbial biomass x Cmic). Select one term and then use it consistently.

Introduction

Line 36: Please, provide a reference.

Line 55: This reference is pretty old. New publications have been made on this topic (e.g. Jílková et al., Domisch et al.). You should refer to newer publications to better define the knowledge gap.

Lines 56-60: Why did you select yeasts as a representative microbial group? This is really not the best alternative. Instead, bacteria and fungi should have been used and described.

Line 57: Define the abbreviatons when first used (abstract stands alone).

Lines 59-60 and 68-69: Why do you think that soil properties should change after honeydew deposition? Provide a background.

Lines 71-72: Why do you think soil mesofauna should change? First mentioned here in the summary. Provide a better background.

Lines76-78: Here you refer to knowledge gaps you want to fill in in your study. However, at the same time you provide references on publications where this has been already investigated. Better formulation and finding knowledge gaps are urgently needed.

Material and methods

Line 93: How was the soil cultivated?

Line 103: The section only includes soil sampling and processing. Therefore delete “and biochemical analyses“ from the title.

Line 124: How deep and what volume of the soil was sampled?

Line 127: Usually, soil is sieved through a 2-mm screen. Why did you select a 5-mm screen?

Line 129: It is unclear to me why did you air dry the samples for enzyme analyses. It seems to be a non-sense to determined enzyme activity in a dry sample. Moreover, even if rewetted, enzyme activities must have been changed. Why did you perform some analyses on the dry and some on the frozen samples? Unclear.

Line 131: It is also a nonsense to determine soil respiration in a frozen sample. Microbial cells must have ruptured due to low temperatures which must have increased available nutrient contents and thus microbial activity and biomass.

Line 167 and 187: What does it mean “oven-dried“? Usually dry weight is determined after drying at 105 °C for 12 h.

Results

Line 259: The title should state: Soil chemical properties.

Lines numbers missing: Provide a table for the data (mean±SE) before planting. In this state, it is hardly understandable. Moreover, data on biological properties are missing at all.

I miss comments on soil temperature and moisture. I do not understand why did you measure those at all as the rationale is missing.

Line 267 and Figure 2: Usually, fauna abundance is expressed per m2, or cm3, or g soil. Not per 125 m3. This makes further comparisons of your data with other studies hard.

Line 273: You cannot state that as you did not test the time factor!

Line 274: Any explanation on this result in the discussion?

Line 277: Not true, the difference was significant only in year 1.

Line 278: Not true! Black sooty mould spots were similar to control.

Lines 282-284: Restructure the sentence so that it is more comprehensible.

Line 298: Microbial response in what? Activity?

Lines 299-300: Delete this sentence.

Lines 314 and 319: Figure A2a and Figure A2b?? Where are they presented? Probably you meant Figure S2a and figure S2b?

Discussion

Improve the discussion section with newer literature.

Line 341: It is not true that total C was not affected.

Line 352: Nous??

Line 382: Why should there be any negative effect??

Reviewer 2 Report

The manuscript is well written and clear in all parts. THere are only some small errors indicated in the attached pdf. THus, only minor revisions are needed.
